# Repetitive sensorimotor mu-alpha phase-targeted afferent stimulation produces no phase-dependent plasticity related changes in somatosensory evoked potentials or sensory thresholds

Steven Pillen[1,2,3,4], Anastasia Shulga[5,6], Christoph Zrenner [1,2,7,8], Ulf Ziemann[1,2], Til Ole Bergmann[1,2,3,4] *

1 Department of Neurology & Stroke, Eberhard Karls University of Tübingen, Tübingen, Germany, 2 Hertie Institute for Clinical Brain Research, Eberhard Karls University of Tübingen, Tübingen, Germany, 3 Neuroimaging Center, Focus Program Translational Neuroscience, Johannes Gutenberg University Medical Center, Mainz, Germany, 4 Leibniz Institute for Resilience Research, Mainz, Germany, 5 Ward for Demanding Rehabilitation, Helsinki University Hospital, Department of Physical and Rehabilitation Medicine, Helsinki, Finland, 6 BioMag Laboratory, Helsinki University Hospital Medical Imaging Center, Helsinki, Finland, 7 Temerty Centre for Therapeutic Brain Intervention, Centre for Addiction and Mental Health, and Department of Psychiatry, University of Toronto, Toronto, ON, Canada, 8 Institute for Biomedical Engineering, University of Toronto, Toronto, ON, Canada

* tobergmann@uni-mainz.de

**Data Availability Statement:** All raw and processed data as well as MATLAB analysis scripts of this project are available for download via the

## Abstract

Phase-dependent plasticity has been proposed as a neurobiological mechanism by which oscillatory phase-amplitude cross-frequency coupling mediates memory process in the brain. Mimicking this mechanism, real-time EEG oscillatory phase-triggered transcranial magnetic stimulation (TMS) has successfully induced LTP-like changes in corticospinal excitability in the human motor cortex. Here we asked whether EEG phase-triggered afferent stimulation alone, if repetitively applied to the peaks, troughs, or random phases of the sensorimotor mu-alpha rhythm, would be sufficient to modulate the strength of thalamocortical synapses as assessed by changes in somatosensory evoked potential (SEP) N20 and P25 amplitudes and sensory thresholds (ST). Specifically, we applied 100 Hz triplets of peripheral electrical stimulation (PES) to the thumb, middle, and little finger of the right hand in pseudorandomized trials, with the afferent input from each finger repetitively and consistently arriving either during the cortical mu-alpha trough or peak or at random phases. No significant changes in SEP amplitudes or ST were observed across the phase-dependent PES intervention. We discuss potential limitations of the study and argue that suboptimal stimulation parameter choices rather than a general lack of phase-dependent plasticity in thalamocortical synapses are responsible for this null finding. Future studies should further explore the possibility of phase-dependent sensory stimulation.

Open Science Framework (OSF) under the following link: https://osf.io/t6vdn/.

**Funding:** This work was funded by the Deutsche Forschungsgemeinschaft (DFG, German Research Foundation, grant no. 362546008; dfg.de) to T.O. B., by a travel grant of the Academy of Finland to A. S., by the German Federal Ministry for Economic Affairs and Energy through an EXIST Transfer of Research grant (no. 03EFJBW169; exist.de) to C. Z., and by the European Research Council (ERC Synergy) under the European Union's Horizon 2020 research and innovation programme (ConnectToBrain; grant agreement No. 810377; erc.europa.eu) to U.Z. The funders had no role in study design, data collection and analysis, decision to publish, or preparation of the manuscript.

**Competing interests:** I have read the journal's policy and the authors of this manuscript have the following competing interests: C.Z. holds equity in sync2brain GmbH (Tübingen, Germany), a start-up spin-off company that commercializes the EEG analysis hardware that was used in this study. S.P., A.S., T.O.B., and U.Z. declare that they have no competing financial interests or personal relationships that could have appeared to influence the work reported in this paper.

## Introduction

Neural oscillations orchestrate spontaneous and task-related brain activity and are involved in a plethora of cognitive functions with a particular relevance for learning and memory [1]. Irrespective of the specific mechanisms of their generation, which may involve alternating phases of de- and hyperpolarization as well as GABAergic dis-/inhibition, neuronal oscillations are commonly associated with rhythmic fluctuations in neuronal excitability [2], and the resulting synchronization of sufficiently large neuron ensembles can be detected in the surface electroencephalogram (EEG). From this temporal alternation of low and high excitability states, the basic mechanism of *rhythmic input-output gain modulation* emerges: During the more excitable state there is an increased likelihood of postsynaptic spiking in response to synaptic input and thus a facilitation of signal transmission, whereas during the less excitable state synaptic input is less likely to cause firing and signal transmission is suppressed. For the human primary motor cortex (M1), this principle has been demonstrated repeatedly by a number of recent real-time EEG-triggered transcranial magnetic stimulation (TMS) studies, demonstrating phase-dependent changes in corticospinal excitability via the measurement of motor evoked potentials (MEPs) in contralateral hand muscles. A phase-dependent modulation of MEP amplitude was observed for up- vs. down-states of the sleep slow (<1 Hz) oscillation [3] as well as peaks vs. troughs of the sensorimotor mu-alpha (~10–14 Hz) rhythm [4–10] (but see [11, 12]). Bergmann et al. further showed that MEPs were specifically facilitated during mu-alpha troughs, but no suppression was observed at any phase, thus favoring asymmetrical pulsed facilitation (or disinhibition) over pulsed inhibition in M1 [7].

Notably, the excitability change during slower oscillations also modulates the amplitude of faster oscillations, constituting the phenomenon of *cross-frequency phase-amplitude coupling* (PAC) [13], a mechanism considered crucial for memory formation (e.g., theta-gamma PAC) [1] and consolidation (e.g., slow oscillation-spindle-ripple PAC) [14]. In fact, *phase-dependent plasticity* has been proposed as a general mechanism by which PAC mediates synaptic plasticity [15]: If a synaptic input volley (from the faster oscillation) arrives during a phase of increased excitability (defined by the slower oscillation), $Ca^{2+}$-influx into the apical dendrites is maximized and long-term potentiation (LTP) may occur, while no such effect (or even long-term depression, LTD) may be produced when the input volley arrives during a phase of decreased excitability. There is indeed experimental evidence for phase-dependent plasticity from both theta phase-triggered electrical stimulation in the rodent hippocampus [16, 17] as well as sensorimotor mu-alpha phase-triggered TMS in the human M1 [6]. In the latter study, subthreshold 100 Hz TMS triplets administered repeatedly into the mu-alpha troughs (i.e., the facilitatory phase), led to an LTP-like increase of corticospinal excitability, whereas the same protocol applied to the peaks did not, but rather induced a trend towards LTD-like decreases [6].

In principle, the mechanism of phase-dependent plasticity should also apply to thalamocortical synapses in the human primary somatosensory cortex (S1). To test this assumption, we here repetitively applied real-time EEG-triggered sensorimotor mu-alpha phase-dependent peripheral electrical stimulation (PES) of the fingertips and assessed changes in thalamocortical synaptic strength via the assessment of somatosensory evoked potentials (SEP) and sensory thresholds (ST).

## Materials and methods

### Participants

16 healthy, right-handed volunteers (mean age ± SD: 26.7 ± 3.8 years, range 21–35, 9 females), without a history of psychiatric or neurological disease, and free of medication, were recruited

via university emails lists and flyers between November 2018 and January 2019 and participated after providing written informed consent. Only the first author of the study (S.P.) had access to any information that could identify individual participants during or after data collection, and those data were kept separate from the pseudonymized experimental data. The study protocol conformed to the Declaration of Helsinki and was approved by the local ethics committee of the University Hospital Tübingen (Ethics Vote No. 1441/2017B01). Inclusion criteria further comprised a right-handedness laterality score > 70% (mean laterality score ± SD: 89 ± 11%, range 77.8–100) according to the Edinburgh Inventory [18] and the presence of a clear sensorimotor mu-alpha peak in the resting-state EEG power spectrum (see description of screening for details).

## Experimental design and procedures

Subjects participated in a screening session, directly followed by preparatory measurements, and a single experimental session (Fig 1). We used real-time EEG phase-triggered PES to repeatedly produce afferent thalamocortical input to cortical neurons in S1 during either (i) peak, (iii) trough, or (iii) random phases of the sensorimotor mu-alpha rhythm. While all phase conditions were intermingled in pseudorandomized order within a single session, for each participant a phase condition was consistently associated with PES to a different finger (i.e., digit I, III, and V; as their topographical representations in S1 are maximally separated [19], but phase-finger allocations were balanced across participants. The rationale was based on the notion of phase-dependent plasticity [6, 15], assuming that repeated afferent input arriving during a phase of relatively increased cortical excitability should result in an LTP-like strengthening of the corresponding thalamocortical synapses in S1, whereas the same input

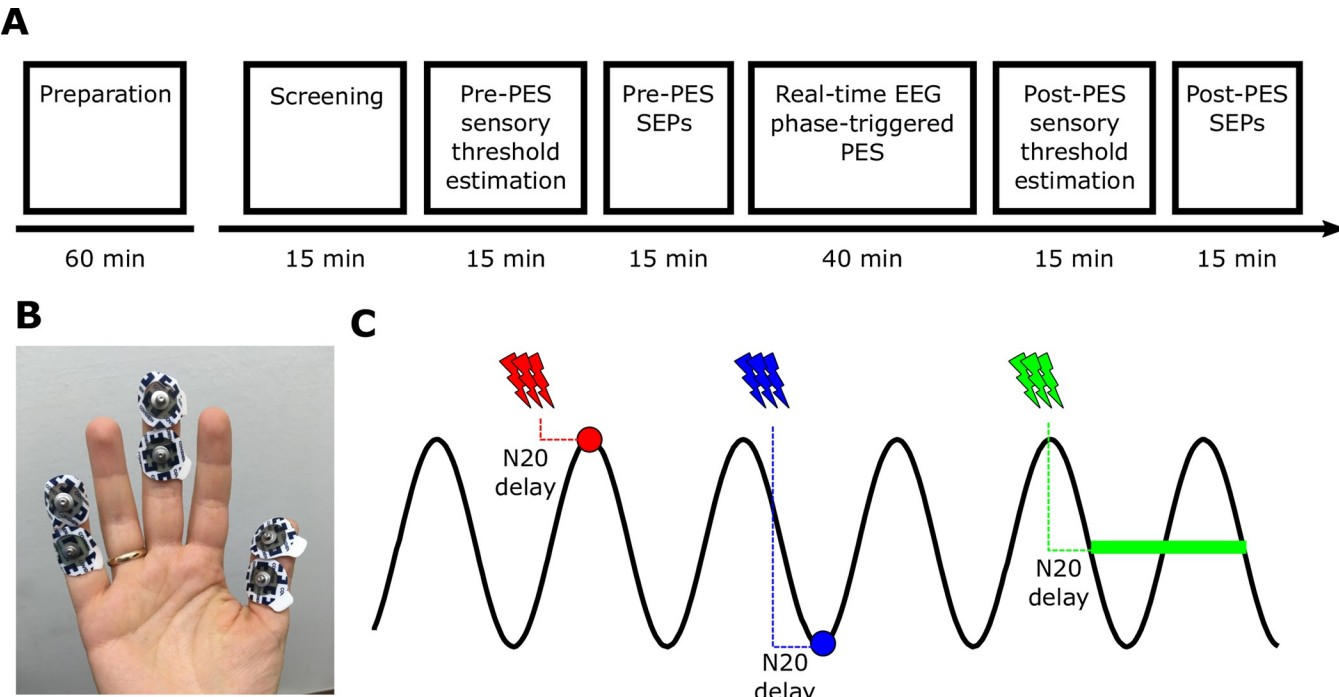

**Fig 1. Experimental design and procedures. (A)** Timeline of sessions and procedures for a single experiment (note that durations provided refer to the time spent on each particular experimental stage and not just the mere measurement time). **(B)** Position of surface electrode pairs for finger-specific PES on the digit I, III, and V of the right hand. **(C)** Schematic depiction of phase-targeted PES, resulting in the arrival of somatosensory input in S1 at the peak, trough, or random phases of the sensorimotor mu-alpha rhythm. PES was applied at an interval prior to the respective phase that reflected the individual N20 latency of the SEP.

repeatedly arriving during phases of relatively decreased excitability should result in LTD-like weakening of the same synapses, both relative to a control condition of phase-randomized targeting for which no plastic effect was assumed. While MEP results from EEG-triggered TMS suggest a phasic facilitation of corticospinal excitability during the mu-alpha trough but no change during the peak [7, 20], it is still unclear whether the same or possibly even the opposite direction of effects holds for the cortical excitability of S1. We thus only formulated the undirected hypothesis of a differential effect of peak- and trough-triggered PES, and a difference relative to the random condition for either one or both of them. We assessed phase-specific changes in the strength of thalamocortical synapses in S1 from before (pre) to after (post) the intervention with respect to their consequences on two outcome variables: (i) the amplitude of the N20 component of the finger-specific SEP, reflecting the first cortical response in S1 [21], which should increase with strengthened synapses, and (ii) the sensory thresholds (ST) of the respective fingers, which should decrease with strengthened synapses.

## Screening session

After screening for right-handedness using the Edinburgh Inventory [18], the EEG cap was prepared and a 3-min eyes-open resting-state EEG was recorded to determine whether the participant's sensorimotor cortex expressed a mu-alpha rhythm of sufficient signal-to-noise-ratio (SNR) to allow robust real-time phase-targeting. We accepted subjects with the presence of a distinct spectral peak in the power spectrum (FFT with non-overlapping 2 s Hanning windows and 0.5 Hz frequency bins) between 8 and 14 Hz with an amplitude of at least twice the background 1/f noise. A rough 1/f correction was applied through multiplying the power value for each frequency bin of the FFT by their respective frequency. Moreover, the individual mu-alpha peak frequency was extracted ($10.8 \pm 1.2$ Hz, mean $\pm$ SD) to individualize the bandpass-filter for the real-time EEG phase-triggered PES intervention.

## Sensory threshold determination

Sensory thresholds (ST) were determined per finger using an adaptive staircasing algorithm (Parameter Estimation by Sequential Testing, PEST [22]) implemented in MATLAB. For this purpose, we applied 40 PES trials per finger via three constant current stimulators delivering single 200 μs positive square wave pulses via surface electrodes (for technical details see PES section below). The thresholds for all three fingers were determined within the same session by three separate PEST algorithms running simultaneously, probing the three fingers in intermingled trials that were pseudorandomized in order so the same finger would never be probed more than twice in sequence and only one finger would be stimulated at a time (thus comparable to the intervention). Participants were unaware of when and to which finger the next stimulus would occur. They were asked to respond verbally whenever they detected a stimulus and to indicate in which finger. The experimenter would enter a response when and if the subject verbally identified a finger. Stimuli for which the location was correctly identified within 1 s were considered 'hits', whereas incorrectly identified fingers or the absence of a response were considered 'misses'. Intensity changes were indicated after each trial by the algorithm and performed manually by the experimenter via the respective turning knob of the device. After completion of 40 trials per finger (120 trials in total), the finger-specific sensory thresholds (ST) were derived as the average of intensities for the last 10 trials of that finger ($2.3 \pm 0.7$ mA (digit I), $1.97 \pm 0.47$ mA (digit III), $1.62 \pm 0.49$ mA (digit V) and later used to individualize the stimulation intensity for both the SEP measurements (at 2.5x ST) and the PES intervention (at 1x ST). The thresholding procedure lasted approximately 15 min and was performed before ($ST_{pre}$) and after ($ST_{post}$) the intervention.

For each finger, the PEST staircasing algorithm started at a stimulus intensity of 1 mA, with a step size of 0.4 mA. If the subject could detect a sensation in the finger, the intensity was reduced by the step size; otherwise, the intensity was increased by the step size. Every time the step direction was reversed, i.e., when the subject went from not detecting a stimulus in the previous round to detecting a stimulus in the current round, or vice versa, the step size was halved. The second consecutive step in the same direction would keep the same step size. The fourth and subsequent steps in a given direction would be double their predecessors; the third step would also be double its predecessor if the step prior to the most recent reversal also involved a doubling. Otherwise, the third step's size would remain the same as the 1st and 2nd in the same direction. In all cases, a minimum step size of 0.01 mA, a maximum step size of 1.00 mA, and a minimum stimulation intensity of 0.2 mA were employed to constrain the PEST algorithm to reasonable values.

## SEP measurement

Finger-specific SEPs were evoked using PES via bipolar surface electrodes at 2.5x the finger-specific ST as identified in the $ST_{pre}$ session. Stimuli consisted of a single square pulses of 200 µs duration and occurred at a randomly jittered interstimulus interval (ISI) of 200–300 ms across fingers (i.e., 200–1500 ms per finger). A total of 250 stimuli were applied to each finger (750 in total) in pseudo-randomized order so that no finger would be stimulated more than twice in sequence and only one finger would be stimulated at a time (thus comparable to the intervention). After completion of the recording session, the average time series for the classical CP3-Fz bipolar montage [23] was calculated per finger and the individual N20 latency was determined by visually identifying the local minimum of the stereotypical negative peak following the peripheral stimulation. Note that the N20 latency is slightly longer for PES of the distal phalanx of the finger (in this experiment: 25.2 ± 1.9 ms) as compared to the median nerve at the wrist. Since no obvious differences between fingers were observed, the average SEP N20 latency rounded to the nearest integer ms was used to individualize the stimulation delay for the intervention (see next section for details). The SEP measurement itself lasted approximately 5 min and was performed before ($SEP_{pre}$) and after ($SEP_{post}$) the EEG phase-triggered PES intervention.

## EEG phase-triggered PES intervention

During the EEG phase-triggered PES intervention, 400 PES triplets (3 pulses at 100 Hz) were applied at 1x the finger-specific ST as identified in the $ST_{pre}$ session to each of the three fingers (1200 trials in total), with all stimuli occurring in pseudo-randomized order so that no finger would be stimulated twice in sequence and only one finger would be stimulated at a time. The assignment of phase-targeting conditions ('peak', 'trough', or 'random') to fingers (digit I, III, or V) was balanced across subjects. Using real-time EEG analysis via the Simulink-based boss-device (sync2brain, Germany), the phase of the ongoing mu-alpha rhythm was estimated online from a signal extracted using a 5-channel C3-centered surface Laplacian montage (aka *Hjorth* montage [24]), that had been repeatedly shown to be predictive of motor corticospinal excitability [4–8, 20]. For a more detailed description of the real-time processing pipeline see the respective section below. The implemented autoregressive forward prediction algorithm [25] allowed us to forecast the phase signal for a time point located at an adjustable interval after the present time point (here: the individual SEP N20 latency). Thereby, PES could be delivered early enough for the resulting afferent input to reach S1 during the desired target phase (peak or trough), effectively correcting for the required conduction time from finger to cortex. The middle of the three stimuli of a 100 Hz triplet was centered on the target phase to

create maximal overlap of the PES triplet with the targeted excitability state. In the random phase condition, PES was applied independent of the mu-alpha phase, i.e., with an additional randomly jittered delay between zero and one full period of the individual mu-alpha peak frequency. That is, only when the phase criterion was met for a given trial, a PES triplet was triggered. The 500 ms post-PES were excluded from the moving window buffer, and a minimum ISI randomly drawn from the range of 1.5 to 2.5 s was added to prevent PES-evoked SEPs and stimulation artifacts in the EEG to compromise phase estimates. The actual ISI then varied additionally depending on when the specific target phase of the current trial could be identified after that minimum ISI and was on average $2.1 \pm 0.02$ [0.313] s (mean $\pm$ SD across subjects [average within-subject SD across trials]) relative to the last PES of any condition when pooled over phase conditions, and $2.1 \pm 0.05$ [0.312] s (peak), $2.1 \pm 0.05$ [0.314] s (trough), $2.1 \pm 0.04$ [0.312] s (random) per condition. Note that the actual ISI between stimulations of the same phase target (and thus same finger) was considerably longer, namely on average $6.3 \pm 0.07$ [2.48] s (mean $\pm$ SD across subjects [average within-subject SD across trials]) relative to the last PES of the same condition when averaged across phase conditions, and $6.3 \pm 0.07$ [2.48] s (peak), $6.3 \pm 0.07$ [2.48] s (trough), $6.3 \pm 0.07$ [2.48] s (random) per condition. The intervention took $40.0 \pm 6.7$ min in total.

## EEG recordings

EEG recordings were performed using a TMS-compatible 80-channel EEG system (NeurOne Tesla, Bittium Finland) in conjunction with a 64-channel EEG cap with Ag/AgCl electrodes (EasyCap, Germany) in a customized configuration with dense electrode arrangement around the motor cortex according to 10–5 EEG labeling system (Fp1, Fpz, Fp2, Afz, F4, F3, F7, F8, Fz, FFC1h, FFC2h, FFC3h, FFC4h, FFC5h, FFC6h, FCz, FC1, FC2, FC3, FC4, FC5, FC6, FCC1h, FCC2h, FCC3h, FCC4h, FCC5h, FCC6h, Cz, C1, C2, C3, C4, C5, C6, CCP1h, CCP2h, CCP3h, CCP4h, CCP5h, CCP6h, CPz, CP1, CP2, CP3, CP4, CP5, CP6, CPP1h, CPP2h, CPP3h, CPP4h, CPP5h, CPP6h, Pz, P3, P4, P7, P8, O1, O2, T7, T8, TP9, TP10). Electrode impedance was kept below 5 kOhm by means of careful preparation of the skin with isopropyl alcohol and abrasive gel (Nuprep, Weaver & Co, USA) before application of conductive gel (GE-Medical Electrode Cream). EEG data were recorded in DC mode with a 1250 Hz anti-aliasing low-pass filter and digitized at 5 kHz with 24-bit resolution. Recording reference was FCz, and the ground electrode was placed on POz.

## Peripheral electrical stimulation (PES)

PES of the three fingers was applied via three separately controlled constant current stimulators (DS7A, Digitimer Ltd, UK), with each DS7A being connected to one finger (digits I, III, and IV). For this, two disposable adhesive electrodes (COVIDIEN Mini Adhesive Electrodes, 24 mm diameter) were placed in bipolar montage on each finger (anode on fingertip and cathode on the palmar middle phalanx; see **Fig 1B**).

## Real-time EEG-triggered stimulation

The EEG-triggered PES intervention was driven by a prototype of the bossdevice (sync2brain, Germany), a Simulink Real-Time (R2017b, MathWorks) based computer system capable of handling real-time biosignal data and generating a trigger signal when a predefined phase is detected in a specific frequency band for a specific electrode montage (cf. [6]). It was set up to receive the real-time digitized output from the NeurOne Tesla EEG system and used in all EEG measurements throughout the study. For phase detection, EEG signals were downsampled to 500 Hz after applying an 80 Hz two-pass (zero-phase) finite impulse response

(FIR) low-pass filter (order 100), and a target signal reflecting the sensorimotor mu-alpha rhythm was extracted via a spatial filter resembling an extended Hjorth-style surface Laplacian montage (i.e., C3 referenced against the average of FC1, FC5, CP1, and CP5). The last 250 samples (500 ms) of data in the buffer were band-pass filtered with a two-pass (zero-phase) finite impulse response (FIR) filter with order 70 and a pass-band of the individual mu-alpha frequency ± 2 Hz. Afterwards, 35 samples of filter edge artifacts on either side of the buffer were removed, and a Yule-Walker autoregressive model (order 15) was used to forward predict the signal for 70 samples (140 ms) based on the remaining 180 samples [26, 27], thus providing ± 70 ms of forecasted data around "time zero" (i.e., "now"). A Hilbert transform converted the band-pass filtered time series into phase information, and the phase value was extracted from the forecasted signal at a certain offset after "time zero" because the PES evoked afferent signal requires time to reach S1 and therefore PES has to be triggered in time before the targeted mu-alpha phase. Specifically, the offset consisted of (1) the individual N20 latency, (2) a 12 ms hardware circuit loop delay compensation (empirically determined and including the effects of EEG hardware anti-aliasing low-pass filters), and an additional 10 ms delay ensuring that the second (middle) pulse of the 100 Hz PES triplet would align with the desired mu-alpha phase. The PES triplet was triggered when the discrete phase value extracted after that offset fell into a phase bin (width: ± pi/50 radians = ± 3.6˚ phase angle) around the current trial's target phase, which was either 0˚ (*peak*), 180˚ (*trough*), or a random phase (*random*). While most of our previous experiments also applied mu-alpha power criteria to target a specific range of the individual mu-power distribution and prevent the mere targeting of band-pass filtered noise (cf. [6, 7, 20]), a technical error prevented power criteria from taking effect in the current experiment, so that the full range of mu-alpha power values were targeted in a phase-specific manner.

## Offline EEG analyses

EEG data preprocessing and analysis were performed on MATLAB 2017b using customized scripts as well as the FieldTrip open-source toolbox (https://www.fieldtriptoolbox.org [28]).

**EEG data preprocessing.** In both sets of EEG recordings, SEP measurements and EEG-triggered PES intervention, PES produced a consistent and very short spike artifact, which was removed by linear interpolation of the affected time intervals from -2 to 2 ms relative to each PES pulse, an interval length identified as adequate by visual inspection. EEG data were downsampled from 5 kHz to 1 kHz and then bandpass-filtered at 0.1–200 Hz using a 3rd order Butterworth IIR filter.

**SEP analysis.** SEP data were segmented into 1 s epochs from -500 to 500 ms relative to each PES pulse. Bad channels were removed and topographically interpolated from their neighbors (channels interpolated per subject: 4.1 ± 1.3, mean ± SD). Then, EEG data were re-referenced to the common average of all channels. Next, pre- and post-interventional SEP data were concatenated into a single time series and subjected to an ICA (RUNICA implementation in FieldTrip). Only components containing SEP information (i.e., resembling a typical N20-P25 waveform and a corresponding topography) were kept and projected back to channel space, while all other components were removed (removed components per subject: 59.57 ± 0.97, mean ± SD). Bad trials were removed via FieldTrip's semi-automated rejection tool ('*summary*' method of *ft_rejectvisual()*, with a z-score of 10 as cutoff value; percentage of removed epochs per subject: 1.9 ± 1.5%, mean ± SD). Then, cleaned epochs were baseline-corrected by subtracting for each trial the respective average signal from -50 to -10 ms pre-stimulus, and the average was calculated across trials separately for each phase condition and for the measurements before (SEP_pre) and after (SEP_post) the EEG phase-triggered PES intervention. The following procedure was used to extract the N20 SEP component: After determining the

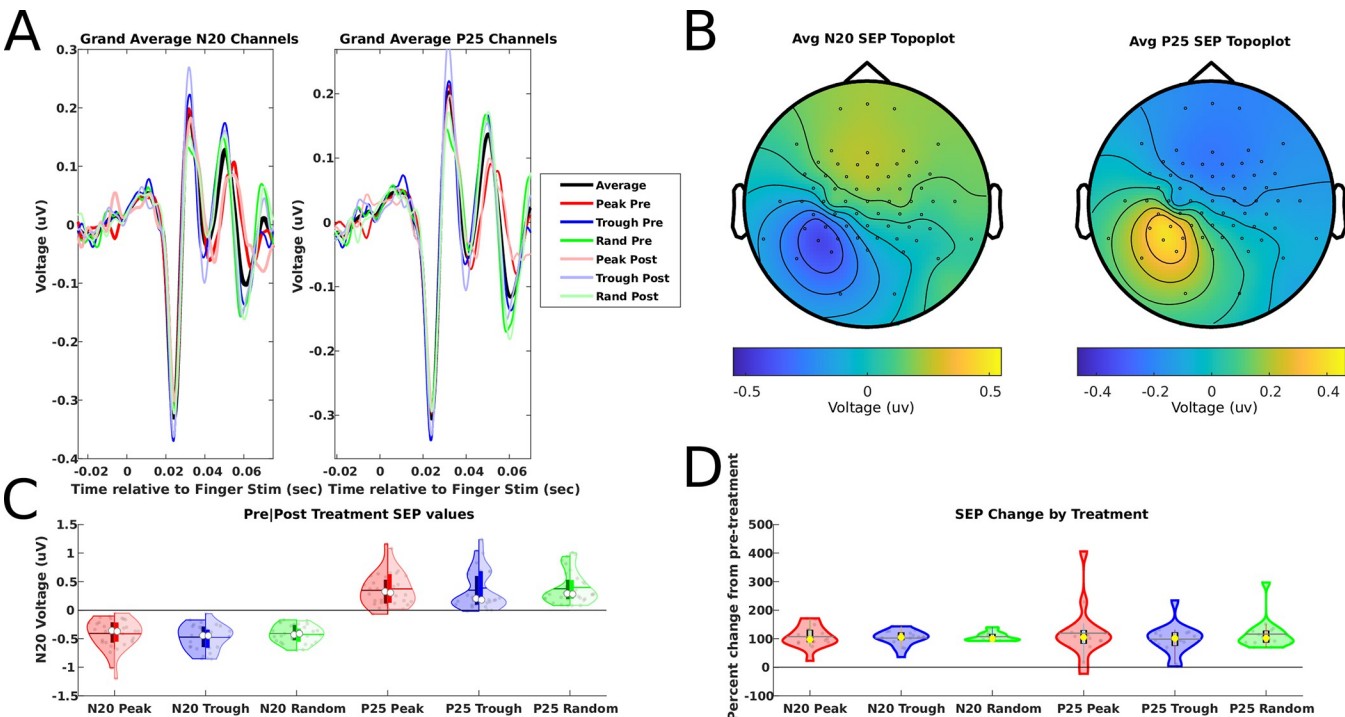

**Fig 2. Somatosensory evoked potentials (SEP).** **(A)** The grand average SEP waveform across all subjects (n = 16), extracted from the N20 and P25 clusters, respectively. **(B)** Grand average topographies calculated from the individual time-locked averages time series +/- 2 ms with respect to each individual subject's N20 and P25 peak, respectively. **(C)** Comparisons of N20 and P25 voltage values pre- and post-intervention, where the left-hand side of the violins represent the pre-intervention and the right-hand side the post-intervention for a given condition. **(D)** Comparisons of the N20 and P25 amplitudes before (SEP$_{pre}$) vs. after (SEP$_{post}$) the EEG-triggered PES intervention expressed in percent change of SEP$_{post}$ from SEP$_{pre}$. For both C and D, the outer shape of the violin represents the kernel density estimation, and gray dots within the violins represent individual subject averages per condition. Within each violin plot a box plot is provided (marking with its upper and lower border the 25% and 75% percentiles), with white circles representing the condition median, horizontal gray lines representing the condition mean, and vertical bars extending from there an additional 1.5 times the interquartile range.

search windows for the individual N20 latency from the grand average SEP (across all subjects, conditions, and measurements) from the classical CP3-Fz montage [23] (**Fig 2A**), the individual SEP N20 component was extracted from the cluster of electrodes surrounding the electrode with the largest negative voltage value within the left hemisphere (which was CP3 in all but two subjects, and CCP5h and Pz in the remaining two). Only the directly surrounding electrodes were included in the cluster, as identified using FieldTrip's "triangulation" method (*ft_prepare_neigbours()*), and time series were averaged across all channels within that cluster. For topographical maps of the SEP N20 component see **Fig 2B**. The individual N20 latency was determined from that signal by finding the local minimum value between 20 and 30 ms post-stimulus, which was also visually inspected to verify that the point in the time series corresponded to the visual appearance of a typical N20 (note that the N20 resulting from stimulation of the fingertips is typically later than the one from median nerve stimulation at the wrist). For each subject, phase condition, and measurement, the N20 amplitude value was determined as the cluster average of absolute voltage values at the individual N20 latency. For the P25 component, an analogous procedure was used, this time focusing on the local maximum within 10 ms following the N20 and then calculating the P25 amplitude from the absolute voltage values across the P25 cluster electrodes at the individual P25 latency. In addition, N20 and P25 amplitudes were also extracted from the classical CP3-Fz montage after minimal ICA cleaning (only removing obvious eye blink/movement components; removed components per subject: 1.53 ± 0.62, mean ± SD).

*Targeting validation for EEG phase-triggered PES intervention.* EEG data from the phase-triggered PES intervention was analyzed post-hoc to verify successful phase targeting for the three experimental conditions. Data were epoched from -500 to 500 ms relative to the estimated time of arrival of the respective afferent signal in S1 (i.e., the time point of the center pulse of the PES triplet plus the individual N20 latency). Epochs were subjected to an ICA (RUNICA implementation in FieldTrip) to remove components containing eyeblinks, muscle noise, and other artifacts that could be discretely contained within a component (number of removed components per subject: 3.7 ± 1.4, mean ± SD), and the C3-centered Laplacian signal was afterwards calculated from the cleaned signal. After band-pass filtering (Hamming windowed FIR filter implemented in EEGLAB's "pop_eegfiltnew" function) the signal at ±2 Hz from the individual mu-alpha peak frequency identified in the initial rs-EEG recording, a Hilbert transformation was applied to each epoch to calculate the phase of the mu-alpha rhythm at each time-point. Since phase estimation at the time of stimulation was corrupted by the interpolated PES artifact and the subsequent SEPs, the phase angle from the previous mu-alpha cycle was taken as a surrogate. To determine the corresponding time point for phase extraction in the previous cycle as accurately as possible, we calculated the period of the subject's individual mu-alpha frequency from the trial-wise pre-PES interval itself (instead of the separately recorded rs-EEG) by detecting the local peaks and troughs in the subject-wise peak- and trough-locked time-series averages and calculating the averaged peak-peak and trough-trough distance (same approach as in [7]). The trial-wise phase values extracted from these time points were then averaged across all trials separately per phase targeting condition. To demonstrate that PES was on average applied to periods of adequate mu-alpha power despite the lack of effective real-time power criteria and that power did not systematically differ between phase conditions, FFTs were conducted separately for all phase conditions based on the -500 to -50 ms relative to the center pulse of the PES triplet (Hanning taper; epoch zero-padded to 2 s to artificially achieve 0.5 Hz frequency resolution).

## Statistics

All statistical analyses were all performed using the statistical software package JASP (version 0.16; JASP Team 2021). To assess the effects of the EEG phase-triggered PES intervention on the dependent variables, two-way repeated-measures ANOVAs (rmANOVA) were conducted with the within-subject factors *Phase* (3 levels: peak, trough, and random) and Time (2 levels: pre- and post-intervention) separately for ST and SEP data, with Greenhouse-Geisser correction for non-sphericity where required (i.e., following a significant Mauchly test). Conditional on a significant F-test, respective post-hoc contrasts would have been conducted to identify the source of the effect. To reduce inter-individual variance, ST and SEP data were also expressed as individual percent change of post- from the pre-interventional measurement (i.e., (post-pre)/100*100), and evaluated by one-way rmANOVAs with the factor Phase only. To identify potential confounders, separate one-way rmANOVAs for the factor Phase were further conducted for the inter-stimulus interval (ISI) and mu-alpha power during the EEG-triggered PES intervention. P-values < 0.05 were considered significant, and data for inferential statistics are reported as mean ± SEM if not stated otherwise. In addition to the classical frequentist's approach, we provide the Bayes Factor (BF), which allows for a more robust and sample size independent quantitative assessment of evidence for the null and alternative hypothesis, respectively. For nonsignificant tests we report $BF_{01}$ to quantify strength of evidence supporting the null hypothesis (H0) and for significant tests we report $BF_{10}$ (i.e., $1/BF_{01}$) to quantify strength of evidence supporting the alternative hypothesis (H1). According to Jeffreys [29], a BF of 1–3 reflects "anecdotal evidence"; 3–10, "substantial evidence"; 10–30,

"strong evidence"; 30–100, "very strong evidence"; and 100, "decisive evidence" for the H0 ($BF_{01}$) and H1 ($BF_{10}$), respectively.

## Results

### No effects of EEG phase-triggered PES on SEP amplitude

Typical SEP waveforms (Fig 2A) and topographies (Fig 2B) were observed during $SEP_{pre}$ and $SEP_{post}$ measurements in all Phase conditions, and SEP amplitude values were evaluated both as absolute values (Fig 2C) and as percent change from pre to post intervention (Fig 2D). The two-way Phase x Time rmANOVA on absolute SEP N20 values revealed no significant main effects of Phase ($F_{2,30} = 0.74$, $p = 0.47$, $BF_{01} = 2.67$) or Time ($F_{1,15} = 0.61$, $p = 0.45$, $BF_{01} = 4.52$) or their interaction ($F_{1.5,27.1} = 0.86$, $p = 0.86$, $BF_{01} = 6.77$), and also the respective one-way Phase rmANOVA on the percent change SEP values revealed no main effect of Phase ($F_{1.3,20.0} = 0.16$, $p = 0.76$, $BF_{01} = 5.56$). Comparable results were obtained when conducting the same rmANOVAs instead based on the SEP P25 values. The analyses for absolute SEP P25 values also revealed no significant main effect of Phase ($F_{2,30} = 0.17$, $p = 0.84$, $BF_{01} = 7.72$) nor a Phase x Time interaction ($F_{2,30} = 0.12$, $p = 0.89$, $BF_{01} = 5.69$), but this time a general increase in P25 amplitudes from pre to post measurements caused a significant main effect of Time ($F_{1,15} = 6.42$, $p = 0.023$, yet with a $BF_{01} = 2.75$ favoring the null hypothesis). Similarly, the respective one-way Phase rmANOVA on the percent change SEP P25 values revealed no main effect of Phase ($F_{2,30} = 0.43$, $p = 0.65$, $BF_{01} = 4.47$). In addition, SEPs extracted from the classical CP3-Fz montage after minimal ICA cleaning revealed no significant main effects of Phase or Time or their interaction for any of the analyses described above (all $p > 0.2$; $BF_{01} > 2$). In summary, the EEG phase-triggered PES had no significant phase-specific impact on the SEP N20 or P25 amplitudes. Table 1 provides raw and normalized SEP N20 and P25 amplitude values per phase condition.

### No effects of EEG phase-triggered PES on sensory thresholds

Sensory thresholds (ST) were evaluated both as absolute values (Fig 3A) and as percent change from pre to post intervention (Fig 3B). The two-way Phase x Time rmANOVA on absolute ST values revealed only a significant main effects of or Time ($F_{1,15} = 25.95$, $p = 0.00013$, $BF_{10} = 2.87$) but not of Phase ($F_{2,30} = 0.84$, $p = 0.44$, $BF_{01} = 2.72$) or their interaction ($F_{1.2,29.9} = 1.04$, $p = 0.34$, $BF_{01} = 5.46$), and also the respective one-way Phase rmANOVA on the percent change ST values revealed no main effect of Phase ($F_{1.3,19.8} = 0.99$, $p = 0.36$, $BF_{01} = 2.94$). In summary, the EEG phase-triggered PES had no significant impact on the STs, besides a general intervention-independent increase in ST over time. Table 2 provides raw and normalized ST values per phase condition.

**Table 1. Mean ± SD for raw (µV) and normalized (% change of post from pre PES-intervention SEPs) SEP N20 and P25 amplitudes for all phase targets (peak, trough, random).**

|  | $SEP_{Pre}$ (µV) | $SEP_{Post}$ (µV) | $SEP_{change}$ (%) |
|---|---|---|---|
| **N20 Peak** | -0.407 ± 0.252 | -0.415 ± 0.276 | 7.010 ± 37.626 |
| **N20 Trough** | -0.473 ± 0.217 | -0.478 ± 0.223 | 2.172 ± 27.062 |
| **N20 Random** | -0.409 ± 0.158 | -0.426 ± 0.147 | 7.045 ± 15.204 |
| **P25 Peak** | 0.347 ± 0.305 | 0.372 ± 0.296 | 19.436 ± 94.094 |
| **P25 Trough** | 0.349 ± 0.319 | 0.386 ± 0.381 | -1.865 ± 52.821 |
| **P25 Random** | 0.375 ± 0.283 | 0.398 ± 0.285 | 15.693± 53.570 |

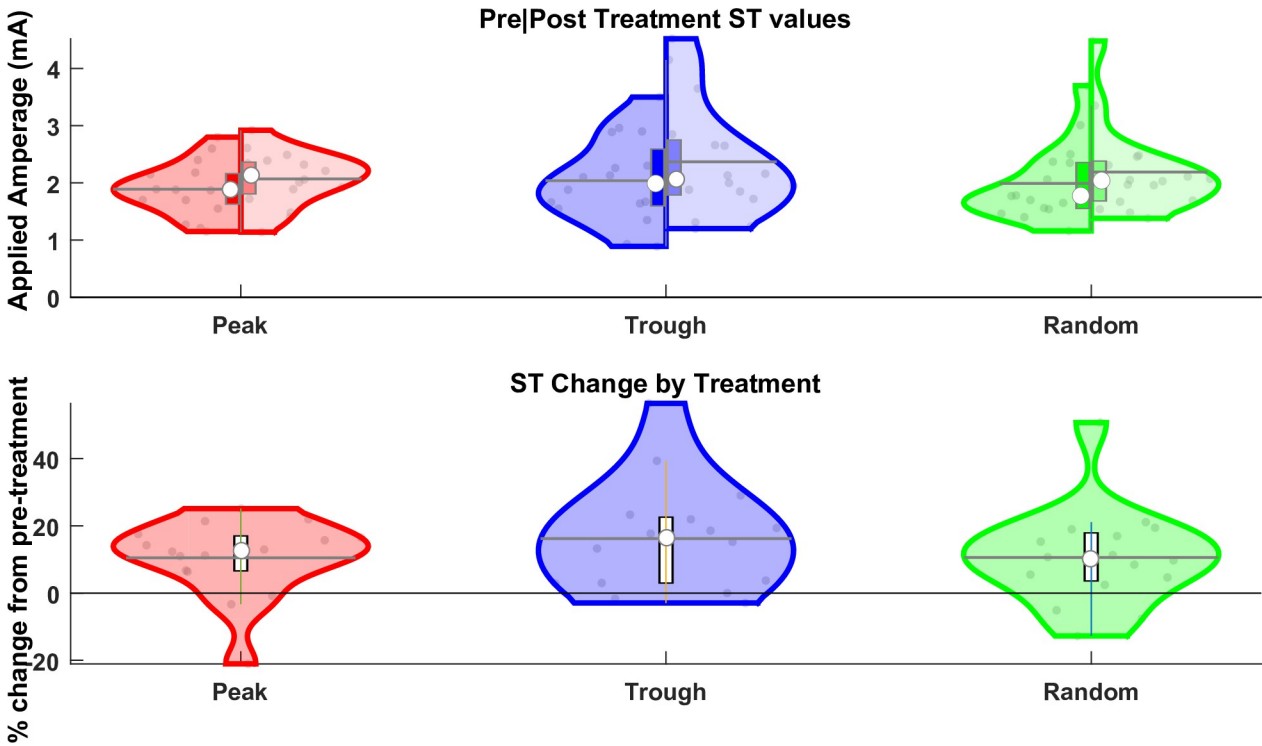

**Fig 3. Sensory Threshold (ST) violin plots. (A)** Split violin graphs where the left-hand side of the violins represent $ST_{pre}$ and the right-hand side $ST_{post}$ for a given Phase condition. **(B)** comparisons of the values after the EEG phase-triggered PES intervention between Phase conditions ($ST_{post}$) expressed in percent change from those before ($ST_{pre}$.) For both A and B, the outer shape of the violin represents the kernel density estimation, and gray dots within the violins represent individual subject averages per condition. Within each violin plot a box plot is provided (marking with its upper and lower border the 25% and 75% percentiles), with white circles representing the condition median, horizontal gray lines representing the condition mean, and vertical bars extending from there an additional 1.5 times the interquartile range.

## Targeting validation for EEG phase-triggered PES intervention

To ensure that the EEG phase-triggered PES intervention worked as intended, targeting mu-alpha oscillations of sufficient power and at the intended phase, respective EEG data were analyzed offline. Time-locked grand averages of the C3-Horth target montage signal already indicate that peak and trough of the mu-alpha oscillation were well targeted, so that several oscillatory cycles are visible in the ~200 ms pre-target period, while averaging canceled out any oscillatory signal due to a lack of phase consistency across trials and subjects for the random phase target as intended (**Fig 4A**). Analyzing the actual targeted phase angle for each trial (as estimated from the preceding mu-alpha cycle to prevent contamination of phase estimates with PES artifacts and SEPs; see methods for details), confirmed good phase targeting across all subjects for both peak (target phase 0˚, average actual phase: 2.1˚; magnitude: 0.99) and trough targets (target phase: 180˚, average actual phase: 179.87˚; magnitude: 0.98) and a

**Table 2. Mean ± SD for raw (mA) and normalized (% change of post from pre PES-intervention STs) sensory thresholds (ST) for all phase targets (peak, trough, random).**

|  | $ST_{Pre}$ (mA) | $ST_{Post}$ (mA) | $ST_{change}$ (%) |
| --- | --- | --- | --- |
| **Peak** | 1.890 ± 0.471 | 2.069 ± 0.470 | 10.480 ± 11.400 |
| **Trough** | 2.038 ± 0.744 | 2.370 ± 0.998 | 16.188 ± 16.095 |
| **Random** | 1.991 ± 0.659 | 2.189 ± 0.776 | 10.601 ± 14.579 |

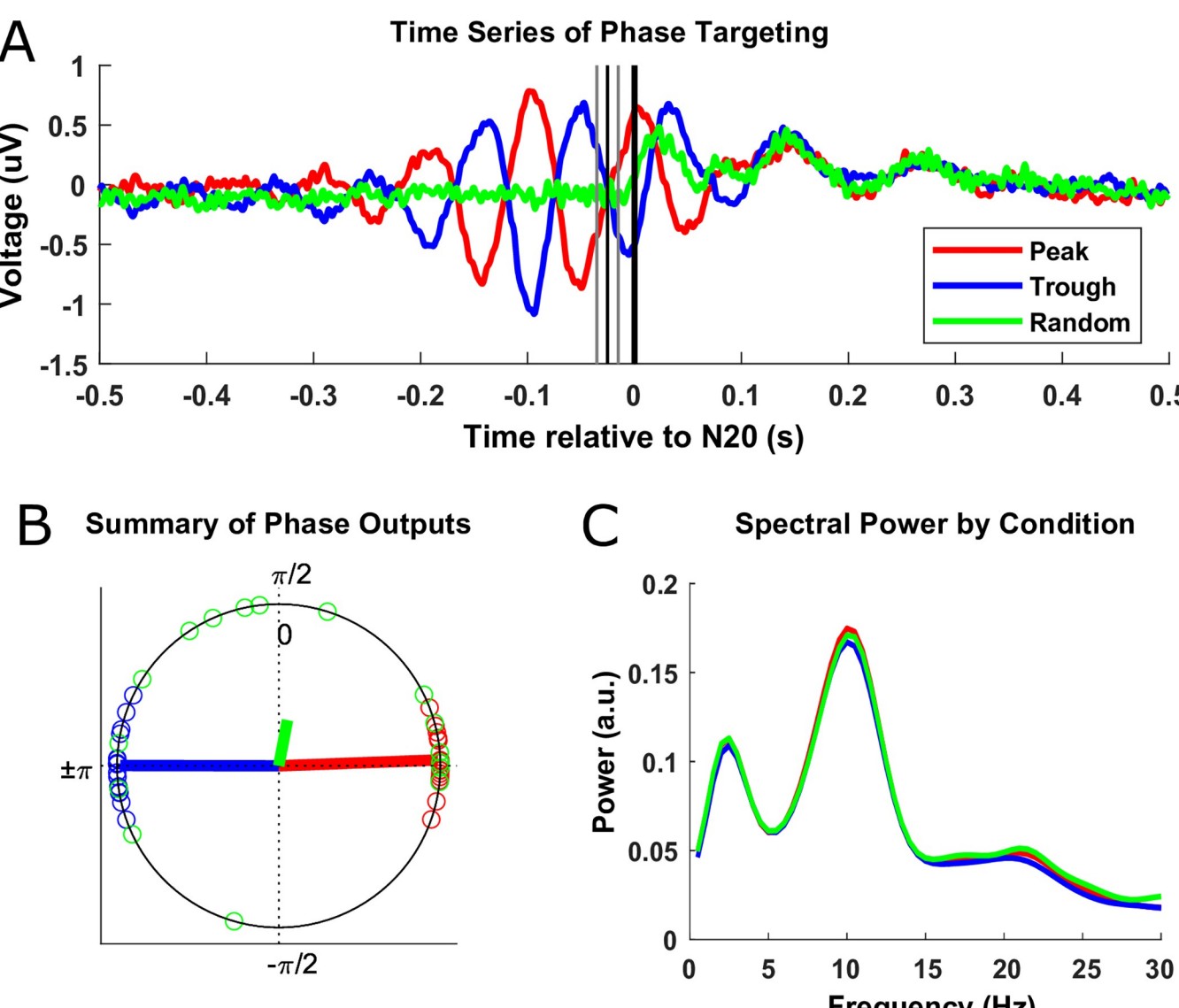

**Fig 4. Targeting validation for EEG phase-triggered PES intervention.** Colors of elements indicate the Phase target conditions peak (red), trough (blue), and random (green). **(A)** Grand average time series across all subjects for the three Phase conditions, individually time-locked to the respectively forecasted phase target (bold vertical line at 0 ms), marking the estimated arrival of afferent input in S1. The average position of the PES triplet, which precedes the phase target by the individual N20 latency and thus varies across subjects, is indicated by the three thinner vertical lines. **(B)** Average phase angles targeted per Phase condition (as estimated from the preceding mu-alpha cycle to avoid corruption by PES artifacts and SEPs, with colored circles on the unitary cycle representing the individual phase averages per target condition and vectors representing the condition-wise grand average phase angles (by their direction) and phase consistencies (by their magnitude) across subjects. **(C)** Grand average power spectra across all subjects for the -500 to -50 ms pre-target (and pre-PES) window for the three Phase conditions.

random distribution for the random phase target (target phase: none, average actual phase: 79.19˚; magnitude: 0.29) (**Fig 4B**). Instantaneous mu-alpha frequencies determined from inter-peak/trough distance (10.12 ± 1.5 Hz, mean ± SD) were slightly lower than those determined via FFT from rs-EEG during the initial screening session (10.8 ± 1.2 Hz, mean ± SD) as revealed by a two-sided paired-sample t-test ($t_{15}$ = 3.06, p = 0.008, $BF_{10}$ = 6.67), confirming that it is preferable to extract individual mu-alpha frequency from the experimental data. Finally, FFTs of the pre-target period revealed clear mu-alpha peaks in the power spectrum

with comparable amplitude across Phase target conditions (**Fig 4C**), as confirmed by an rm-ANOVA revealing no main effect of Phase on pre-PES mu-alpha power $F_{1.3,19.3} = 0.66$, $p = 0.46$, $BF_{01} = 4.05$). Also, ISIs preceding the respective trials did not differ between Phase targets, neither relative to the previous trial of any condition ($F_{2,30} = 0.13$, $p = 0.88$, $BF_{01} = 5.57$) nor to the previous trial of the same condition ($F_{2,30} = 1.07$, $p = 0.36$, $BF_{01} = 3.07$).

## Discussion

We here explored the idea of inducing phase-dependent plasticity in thalamocortical synapses of S1 via repeated EEG phase-triggered electrical stimulation of the fingertips with the afferent input signals timed to reach the cortex during the peak or trough or at random phases (as a control) of the sensorimotor mu-alpha oscillation. If successful, such a paradigm would not only provide experimental means to study the neurophysiological underpinnings of phase-dependent plasticity [15] but may also have implications for neuromodulatory approaches in neurorehabilitation. Unfortunately, we did not observe any phase-specific effects of the intervention, neither on SEP N20 or P25 amplitudes nor sensory thresholds measured before and after the intervention. How is the null finding of this study to be interpreted? There are two possible scenarios. The first possibility is that specific parameter choices made our experimental manipulation unsuitable for inducing phase-dependent plasticity or our dependent measures insensitive for assessing them, or both. The second possibility is that our hypothesis was incorrect, and phase-dependent plasticity does not exist for the specific case of afferent input to thalamocortical synapses in the human S1. We will discuss these possibilities in detail in the following sections as well as the conceptual similarities and differences between phase-dependent and spike-timing dependent plasticity.

### Protocol parameters that may have prevented phase-dependent plasticity

Several aspects of the experimental design or specific parameters may have prevented a successful induction of plasticity-based changes in thalamocortical synapses by our protocol. First, phase targeting may have been suboptimal. While the intended phase angles were successfully targeted, which is not trivial given the additional uncertainty due to the longer forecasting time period necessary for starting the stimulation N20 ms before the actual phase target, it is well possible that the targeted phase angles were not ideal for inducing plastic effects. Studies using EEG-triggered TMS of M1 have repeatedly shown that troughs and peaks of the sensorimotor mu-alpha rhythm are associated with different states of corticospinal excitability [4–10] (but see [11, 12]), which is driven by a rhythmic facilitation during the troughs relative to baseline [7], and phase-dependent plasticity was demonstrated for TMS triplets delivered into the trough [6]. However, the relevant phase angels may differ for S1, the main source of the sensorimotor mu-alpha oscillation, and slight changes in dipole orientation may cause significant shifts in oscillatory phase as assessed on the scalp surface with a C3-centered Hjorth montage [4, 30]. Maybe targeting different phase angles would have resulted in a different outcome. Beside differences in preferred phase for phase-dependent plasticity between M1 and S1, gyral geometry and layer specific effects should be considered. It is possible that the effective phase at relevant thalamocortical synapses in layer 4 pyramidal cells of S1 is shifted relative to the scalp EEG, which is particularly sensitive to signals from superficial layers closest to the scalp but also highly synchronized signals originating from the long apical dendrites of large layer 5 pyramidal cells that are oriented perpendicular to the scalp at the gyral crown [31, 32], with the latter most strongly expressing the rhythms of the alpha band in the monkey [33–36] and likely also in the human brain [37–39]. Therefore, given the assumed location of phase-dependent synaptic plasticity in layer 4, the phase targets may have been

suboptimal as well. However, the use of PES triplets at 100 Hz may have even compensated for slightly missing the optimal phase target. Second, on the other hand, using PES triplets instead of single pulses may have also led to a reduction of temporal precision and prevented the proposed mechanisms of phase-dependent plasticity to take effect. However, given the effective induction of LTP-like plasticity using mu-alpha trough targeted 100 Hz TMS triplets [6] as well as the successful use of high-frequency bursts in a novel paired associative stimulation (PAS) protocol [40, 41], this choice is unlikely responsible for the complete lack of phase-specific after-effects observable in this study. Nonetheless, it should be noted in that previous work using 100 Hz TMS triplets, the first of the three pulses pulse was applied at the target phase angle while the following two pulses already fell into the beginning of the rising phase of the mu-alpha cycle, a phase that was later shown to be even more excitable than the very trough [42]. It is thus possible that a slightly different phase-timing of the triplet would have led to different results in the current study. Third, due to a technical error, no mu-alpha power criteria were effective during real-time phase detection, so that also periods of low mu-alpha power were included where the target signal may have mainly reflected band-pass filtered noise. However, the screening procedure ensured that only subjects with substantial mu-alpha power were included, and the power spectra from the 500 ms pre-PES triplet baseline intervals also revealed considerable mu-alpha power (Fig 4C). In fact, other EEG phase-triggered plasticity protocols have deliberately omitted power criteria to achieve short inter-trail intervals (cf. [43]). However, the unintended inclusion of low power trials may have diluted the phase effects (which are mainly observed for high power trials [10]), and maybe a power threshold would have increased the experimental peak-trough difference and produced significant differences. Fourth, it is possible that the PES intensity was too low. We stimulated at peri-threshold intensity (i.e.,1x the finger-specific ST), which is below the motor threshold for causing muscle responses by PES and below the typical PES intensity of 2-3x the sensory ST used for SEPs or paired associative stimulation (PAS) interventions [44]. This low intensity was chosen to maximize the relevance of background excitability fluctuations and prevent any confounding by PES induced motor responses. However, the afferent input to the thalamocortical synapses may have been too small to produce plastic effects and higher intensity may have been more effective. Fifth, we applied only 400 PES triplets (1200 stimuli) per phase condition/finger during the intervention, which may not be enough in comparison to the 250 single PES stimuli for each finger already during each SEP measurement. Maybe a larger number of PES trials would have been needed to change excitability in thalamocortical synapses. Or, from a different perspective, maybe the many pulses applied during the SEP and ST measurements preceding the intervention did already induce some plastic after-effects that did block the subsequent attempt to induce plastic changes via the phase-triggered PES. Sixth, the interleaving of trials targeting different S1 finger representations at different phase angles within the same session may have disrupted plastic effects via lateral inhibition or other unexpected interactions between the targeted neuronal representations. Seventh, we did not direct the participants' attention towards the stimulated hand as recommended for maximizing the effects of PAS [44], as this would have caused mu-alpha desynchronization (power decreases) and thereby strongly reduced the amplitude of the targeted oscillations. Eights, it is possible that phase-dependent plastic changes were indeed induced somewhere in S1, but not at those neuronal elements that would result in a change of SEP N20 or P25 amplitude or sensory threshold. Alternatively, plasticity induced in relevant thalamic-cortical synapses may have been masked or compensated for by other plastic changes in synapses upstream to the thalamocortical input layer. Ninth, our SEP and ST determination procedures may have been insensitive for the induced effects. Regarding the STs, we observed a general phase-condition unspecific increase over time, which may have been caused by sensory habituation or reduced attention

but also a potential decrease in finger electrode impedance over time (which we did unfortunately not monitor), which may have in turn caused the constant current stimulator to apply lower voltages to achieve the fixed current strength, making the stimuli more difficult to detect. While considered unlikely, even a phase-unspecific general PES-intervention induced long term depression (LTD) is theoretically possible. However, irrespective of its reason, this general decrease cannot explain the lack of differential effects between the phase conditions. Regarding the SEPs, we have analyzed both carefully selected SEP ICA components from subject-specific SEP clusters as well as classical CP3-Fz-montage SEP components following minimalistic ICA cleaning and did not observe any differential effects in either one of them, neither with a frequentist nor a Bayesian statistical approach. While it remains possible that other, more derived EEG indices would have revealed unexpected differences elsewhere, we deliberately refrained from further exploratory analyses to prevent multiple comparison issues. Finally, we did not include a low-level control condition such as "sham" or no-stimulation, i.e., a finger to which no effective PES would be applied in between SEP and ST measures, but only a high-level control condition with random-phase PES application. While a no-stimulation condition may have helped to explain the general ST increases, only the random condition did effectively control for any placebo effects or other nonspecific stimulation effects that are not due to the phase-dependent plasticity mechanism under investigation. In summary, there are many possible reasons why our specific protocol has not produced the expected effects, and the general hypothesis of a sensorimotor mu-alpha phase-dependent plasticity in S1 based on afferent inputs should not yet be discarded without further investigations.

## Plasticity potential of thalamocortical synapses in S1

One the other hand one could argue that thalamocortical signal transmission is too hard-wired and that the respective thalamocortical synapses are not sufficiently susceptible to experience-based plasticity. However, we consider this unlikely for the following reasons. Firstly, during mammalian ontogenetic development, the refinement of topographical projections from the thalamus to S1 is thought to arise through an activity-dependent mechanism in which thalamic axons compete for cortical targets [45]. In fact, thalamocortical synapses may initially be formed as silent synapses and only made functional by subsequent N-methyl-D-aspartate (NMDA) receptor-dependent LTP and LTD, which regulates the efficacy of these functional synapses and contributes to experience-dependent changes in S1 thalamocortical circuits[45, 46]. But could this plasticity of thalamocortical synapses be confined to particular time windows during early development? Studies using sensory learning paradigms show that somatosensation can still be altered in the adult brain, and the exposure to sensory stimulation protocols for minutes to hours can lead to changes in human somatosensation and goal-directed behavior even without training, when stimulation protocols are optimized to alter synaptic transmission and efficacy [47–49]. For example, high-frequency sensory stimulation of the fingertip for 20 min has led to changes in human tactile discrimination performance [50]. In principle, S1 therefore seems to be a valid model for studying afferent stimulation induced plasticity in humans.

## Phase-, state-, and spike-timing dependent plasticity

The idea of phase-dependent plasticity is reminiscent of previously described and successfully demonstrated principles, such as spike-timing dependent plasticity (STDP). Also, the particular protocol used in this study shares certain features with a family of protocols termed paired associative stimulation (PAS). In the following, we will discuss the similarities and differences

between phase-dependent and spike-timing dependent plasticity as well as other state-dependent plasticity protocols.

STDP results in either synaptic long-term potentiation (LTP) or long-term depression (LTD), depending on the timing of the pre- and postsynaptic stimuli [51]. Even though the concepts of STDP and phase-dependent plasticity should not be confused, it is well possible that phase-dependent plasticity partially relies on the principles of STDP. If synaptic input arrives at a time when the postsynaptic neuron is in a state of increased excitability (more excitable oscillatory phase) it becomes more likely that the resulting excitatory postsynaptic potential will pass the firing threshold of that neuron and cause a postsynaptic action potential. Also, the temporal delay between pre- and postsynaptic activity would fulfill the requirements for STDP-based LTP as the former actually caused the latter. In contrast, the same stimulus arriving during a state of decreased postsynaptic excitability (less excitable oscillatory phase) will less likely result in a suprathreshold postsynaptic response and thus not cause STDP-based LTP. However, it also does not fulfill the requirements for STDP-based LTD, as no presynaptic spike arrives explicitly after a postsynaptic spike is fired. Notably, this is in line with the phase-dependent plasticity effects observed with real-time EEG phase-triggered TMS in M1 [6], demonstrating LTP-like plasticity when targeting the more excitable phase the mu-alpha oscillation, but no LTD-like plasticity when targeting the less excitable phase (note that a trend-wise MEP decreases was observed though).

While STDP was originally described in slices and animals, STDP-like plasticity has also been studied in humans using PAS protocols [44, 52]. Sensory PAS repeatedly pairs TMS of the sensorimotor cortex with PES, e.g., of the median nerve at the wrist, to study cortical plasticity in humans [44, 52, 53]. Depending on the relative timing of the two stimuli within each pair, i.e., whether the afferent input arrives at the targeted cortical neurons shortly before or after the TMS pulse, PAS can lead either to LTP-like strengthening or LTD-like weakening of the involved synapses, both at the cortical and spinal cord levels [44, 53]. In most PAS protocols, the two stimuli converge in M1, and LTP- or LTD-like changes in the strength of synapses on corticospinal output neurons are assessed via the resulting changes in MEP amplitude. For M1-PAS, it is assumed that the afferent signal from median nerve stimulation arrives via the sensory thalamus, thalamo-cortical projections to S1, and eventually cortico-cortical projections from S1 to M1. However, PAS can also target S1 directly [44, 54]. For example, median nerve PES coupled to TMS of S1 increased exclusively the amplitude of the P25 component of median nerve PES somatosensory-evoked potential (SEP), which is probably generated in the superficial cortical layers of S1, while SEP components reflecting neuronal activity in deeper cortical layers of S1 (N20 component) or thalamic nuclei (P14 component) remained constant [54]. The authors thus plausibly attributed this finding to STDP-like plasticity of neuronal synapses located in upper cortical layers of S1. In support of that notion, TMS applied to S1 near-synchronously to an afferent signal containing mechanoreceptive information led to SEP changes around 21–31 ms, localized to a tangential source located in Brodmann area 3b, with their timing and polarity suggesting modification of upper cortical layers [55]. Following a rapid-rate PAS of S1-TMS and afferent stimulation, Tsang et al. [56] reported reduced SEP paired-pulse inhibition and a trend-wise increase in both N20 and P25 amplitudes. Moreover, cortico-cortical PAS protocols have been established using dual-coil TMS protocols [57, 58]. In summary, for all these protocols the principle of STDP is explicitly implemented, with PES, sensory stimulation, or TMS generating the presynaptic spike and a TMS pulse generating the postsynaptic spike via suprathreshold stimulation of a neuron somewhere in the respective cortical area.

In addition, there are also a number of protocols that have replaced either of the two PAS components with specific endogenous brain states to produce LTP- or LTD-like effects. For

example, Thabit et al. [59] omitted PES, but instead delivered TMS to M1 during the motor preparation time of a simple reaction time task (and thus during the expected endogenous pre-activation and increased cortical excitability of M1). They found indeed that TMS delivered at 50 ms before or 100 ms after movement onset caused a subsequent increase in MEP amplitude, suggesting LTP-like plasticity in M1. Other studies have omitted TMS, and instead timed the application of PES to specific endogenous states. Mrachacz-Kersting et al. [60] reported LTP-like MEP increases in the tibialis anterior muscle after PES of the common peroneal nerve was repeatedly coupled to endogenous activation of M1 during the peak negativity of the contingent negative variation (CNV) related to imagined movements. Notably, Jochumsen et al. [61] found LTD-like MEP decreases in the same muscle when coupling PES of the tibial nerve (innervating the antagonistic soleus muscle) to the CNV instead. We are not aware of any reports, where endogenous activation of S1 has been associated with external stimulation. Importantly, all these studies do not meet the criteria of STDP in the strict sense, since no two spikes are timed precisely in the order of milliseconds. Instead, these studies are more in line with the notion of gating [62], where an afferent or transcranial stimulus is applied during a more temporally diffuse state of experimentally or spontaneously increased excitability. It is conceivable that phase-dependent plasticity simply reflects a special case of state-dependent plasticity and is merely mediated by the excitability state-related amplification of stimulation-induced postsynaptic potentials. However, it is also possible, that the temporal structure of a neuronal oscillation or even the specific neuronal processes mediating it (e.g., inhibition/disinhibition) are relevant factors determining phase-dependent plasticity effects beyond the mere excitability level at the time of stimulation.

## Conclusion and outlook

In summary, we found no change in SEP N20 or P25 amplitude or sensory thresholds following phase-dependent afferent stimulation of the fingertips repetitively targeting peaks or troughs of the somatosensory mu-alpha rhythm. We argue that this null finding rather reflects suboptimal stimulation parameters than a general lack of phase-dependent plasticity in thalamocortical synapses. Future studies should explore higher-dose stimulation protocols with more stimuli and higher stimulation intensities.

## Author Contributions

**Conceptualization:** Steven Pillen, Anastasia Shulga, Ulf Ziemann, Til Ole Bergmann.

**Data curation:** Steven Pillen.

**Formal analysis:** Steven Pillen.

**Funding acquisition:** Til Ole Bergmann.

**Investigation:** Steven Pillen, Anastasia Shulga.

**Methodology:** Steven Pillen, Christoph Zrenner, Til Ole Bergmann.

**Project administration:** Til Ole Bergmann.

**Software:** Christoph Zrenner.

**Supervision:** Til Ole Bergmann.

**Visualization:** Steven Pillen.

**Writing – original draft:** Steven Pillen, Anastasia Shulga, Til Ole Bergmann.

**Writing – review & editing:** Steven Pillen, Anastasia Shulga, Christoph Zrenner, Ulf Ziemann, Til Ole Bergmann.

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
