## [Decision Letter · Decision Letter 0]

24 Jul 2023

PONE-D-23-15283Repetitive sensorimotor mu-alpha phase-targeted afferent stimulation produces no phase-dependent plasticity related changes in somatosensory evoked potentials or sensory thresholdsPLOS ONE

Dear Dr. Bergmann,

Thank you for submitting your manuscript to PLOS ONE. After careful consideration, we feel that it has merit but does not fully meet PLOS ONE’s publication criteria as it currently stands. Therefore, we invite you to submit a revised version of the manuscript that addresses the points raised during the review process.

The reviewers have shown interest in the results, while also raising serious technical concerns. Please revise the manuscript to provide clarity on the experimental design, the formulation of the hypothesis, and address the technical issues pointed out by the reviewers.

We look forward to receiving your revised manuscript.

Kind regards,

Gennady S. Cymbalyuk, Ph.D.

Academic Editor

PLOS ONE

“I have read the journal's policy and the authors of this manuscript have the following competing interests:

C.Z. holds equity in sync2brain GmbH (Tübingen, Germany), a start-up spin-off company that commercializes the EEG analysis hardware that was used in this study.

S.P., A.S., T.O.B., and U.Z. declare that they have no competing financial interests or personal relationships that could have appeared to influence the work reported in this paper.”

Reviewers' comments:

Reviewer's Responses to Questions

**Comments to the Author**

1. Is the manuscript technically sound, and do the data support the conclusions?

Reviewer #1: Partly

Reviewer #2: Yes

2. Has the statistical analysis been performed appropriately and rigorously? 

Reviewer #1: Yes

Reviewer #2: Yes

3. Have the authors made all data underlying the findings in their manuscript fully available?

Reviewer #1: Yes

Reviewer #2: Yes

4. Is the manuscript presented in an intelligible fashion and written in standard English?

Reviewer #1: Yes

Reviewer #2: Yes

5. Review Comments to the Author

Reviewer #1: Pillen et al report a study testing hypothesis that the sensory cortical mu-alpha rhythm, measured near somatosensory cortex with EEG, reflects fluctuating periods of plasticity in thalamocortical processing streams. Past work has shown mu/alpha phase varying excitability and plasticity in the motor cortex, using transcranial magnitic stimulation (TMS), and this conforms with known theta-phase dependent plasticity in the hippocampus. This study sought to test whether such findings generalize to sensory regions, and to bottom-up processing streams (i.e. thalamocortical inputs). The researchers stimulated subjects’ fingertips with peripheral electrodes in short gamma-bursts (100hz) at different phases of the simultaneously-recorded EEG mu/alpha rhythm (trough, peak, random) and tested how a) sensory evoked potentials (the N20/P25) and b) sensory thresholds changed pre-vs-post stimulation protocol (which lasted 40 minutes). Interestingly, none of the stimulation conditions lead to increased somatoensory evoked potentials (i.e. evoked brain responses to sensory stimulation — N20 or P25), and all three lead to increased sensory threshold, suggesting the absence of phase-dependent plasticity. Thus, the paper on the whole reports negative results. The study nevertheless appears to have been carried out carefully and to a high technical standard. As alternatives to a true null-hypothesis (that phase dependent plasticity does not exist in the bottom-up processing stream for somatosensation), they propose that either the stimulation procedure was inadequate OR that the dependent measures were insensitive. The discussion of these possibilities and the results in general is thoughtful and sound. I would therefore recommend publication after the authors address the following

1. Thalamocortical inputs in primary sensory regions (from primary thalamic nuclei) primarily synapse in deeper layer 4, while the EEG signals and maximal TMS effects (from past studies) might be biased toward layer 1 of the cortex, where inputs from other cortical regions and secondary thalamic nuclei synapse. Perhaps the differences seen in this study compared to past findings of m1 phase-dependent plasticity/excitability reflect not a difference between motor and sensory cortices, but between bottom-up/lay4 vs top-down/intracortical/lay1 synapses.

2. The consideration of stimulation-related caveats is thorough (7 distinct points) but the discussion of limitations in their recording/analysis of the SEPs and STNs is lacking. I can think of several potential caveats -- some of which can be added to the discussion, while some may be directly explored.

-The mu-rhythm was measured using a surface laplacian montage. This would likely bias the recorded Mu to superficial layers, while the relevant mu-alpha should be strongest in granular layer 4 (thalamocortical recipient layer). It has been shown that cortical regions may display distinct alpha-band oscillations, differentially related to attention and memory retention, and with one being potentially thalamocortical and another being corticocortical (see reviews by Klimesch et al as well as e.g. Lopes Da Silva, 1980; Halgren et al. 2019; Sokoliuk et al 2019; Rodriguez-Larios et al eNeuro). These different processes may differ in both frequency and spatial distribution. Further, there may be distinct visual vs somatosensory mu/alpha rhythms, which could both present in all electrodes (separable mainly via ICA) and therefore may have biased/affected the researchers ability to select the relevant mu/alpha rhythm. The chosen 8-14Hz range for identifying individual mu-alpha rhythms in this paper is a broad band, and it’s possible that the researchers selected the wrong oscillation to stimulate with respect to.

-Is it possible that the sensory threshold and SEP tests induced some plasticity, making it more difficult to identify plasticity effects? upwards of 300 stimuli were applied to each finger before the intervention even began. The authors consider that the intervention (of 400 stimulations per finger) were not strong enough relative to the pre/post tests of SEP/ST, but what if the pre/post tests themselves were too strong?

-The ICA-component selection methods seem highly subjective “Only components containing SEP information (i.e., resembling a typical N20-P25 waveform and a corresponding topography) were kept”. A more common approach in the literature is to use ICA only to remove irrelevant components (blinks, eye movements, musculature tension, cardiac signal). Could this metholodogy for isolating N20/P25 wavefoms have been too restrictive, leading to the null result?”

-On the other hand, because only the p20/p25 components were retained after the ICA, could the authors use a global-field power estimate of the ERP amplitude? this might be a more powerful/sensitive metric than simply using the same limited cluster of electrodes across all subjects.

3. Further, was the impedance of the finger electrodes tested before and after the intervention?

4. it is hard to discern different conditions in the line plots in figure 2A, although I understand that this is, to some degree, the point! nevertheless, please spread out the x-axis or make the lines thinner.

5. Authors should discuss the absence of a “no stimulation” or sham condition.

6. I may be confused, but the fact that some INCREASE in sensory threshold was found for all cases is interesting. This is a drop in sensitivity. It is difficult to make sense of this effect in light of their interpretation that their protocol should induce LTP but not induce LTD (which, it may have in all three cases). This effect was strong and clearly present in nearly all subjects (and was statistically significant). Could a change in stimulation electrode impedance/contact/etc account for it? Or changes in the tissue e.g. inflammation? Or perhaps changes in arousal could have lead to drops in skin conductance, which would alter the impact of the stimulating electrode.

7. Another potential explanation for the null results could have been that plasticity in synapses upstream from thalamocortical inputs to S1 masked, blunted, or compensated for thalamocortical plasticity (in a phase-dependent or independent manner). Indeed, as they point out in the discussion, a past study identified only plasticity for concurrent PES and direct cortical TMS stimulation, and this affected mainly the p25, interpreted as a layer 1 effect, while the n20 was not affected (interpreted as a lack of a thalamocortical effect).

Reviewer #2: Thank you for the opportunity to review this manuscript.

The experiment investigates whether (and how) sensorimotor mu-alpha phase alters cortical excitability in sensory cortex. The authors use state of the art real-time EEG phase prediction to trigger peripheral electrical stimulation timed to occur at different parts of the sensorimotor mu oscillation. The goal is to induce phase-dependent plasticity in thalamocortical synapses, as has been shown in corticospinal output from M1. However there were no differences observed between the peripheral pulses timed to occur in peak, trough or random phases.

This is a negative study, however as outlined in the discussion, there are several key limitations to the experimental design that may have undermined the ability to draw any conclusions about the proposed relationship between phase and excitability. Most importantly, it is not clear that the phase estimated at C3 would be the same as phase overlying the sensorimotor cortex. As the authors mention, even a small difference in electrode placement and orientation can cause a significant shift in the measured oscillatory phase. This issue may mean that while the estimation is reliable at C3, the only conclusion that can be drawn is that SEPs delivered according to mu-alpha phase measured at C3 do not affect plasticity in nearby sensory cortex.

All other limitations as discussed by the authors may also contribute to the negative findings. In addition, the choice to have the central pulse of a triplet at the target (i.e., have one pulse just before, one at target and one after target) may also play a role as some evidence suggests that rising or falling phase is important for long term potentiation, rather than peak or trough itself.

Minor points:

The channels listed in figure 2B overlap and are unreadable.

The boxplots with 25-75% ranges within the violin plots are so small that they are unreadable in figure 2 and 3.

6. PLOS authors have the option to publish the peer review history of their article (what does this mean?). If published, this will include your full peer review and any attached files.

Reviewer #1: **Yes: **Jordan P. Hamm

Reviewer #2: No

---

## [Author Response · Author response to Decision Letter 0]

21 Sep 2023

Reviewer #1: Pillen et al report a study testing hypothesis that the sensory cortical mu-alpha rhythm, measured near somatosensory cortex with EEG, reflects fluctuating periods of plasticity in thalamocortical processing streams. Past work has shown mu/alpha phase varying excitability and plasticity in the motor cortex, using transcranial magnitic stimulation (TMS), and this conforms with known theta-phase dependent plasticity in the hippocampus. This study sought to test whether such findings generalize to sensory regions, and to bottom-up processing streams (i.e. thalamocortical inputs). The researchers stimulated subjects’ fingertips with peripheral electrodes in short gamma-bursts (100hz) at different phases of the simultaneously-recorded EEG mu/alpha rhythm (trough, peak, random) and tested how a) sensory evoked potentials (the N20/P25) and b) sensory thresholds changed pre-vs-post stimulation protocol (which lasted 40 minutes). Interestingly, none of the stimulation conditions lead to increased somatoensory evoked potentials (i.e. evoked brain responses to sensory stimulation — N20 or P25), and all three lead to increased sensory threshold, suggesting the absence of phase-dependent plasticity. Thus, the paper on the whole reports negative results. The study nevertheless appears to have been carried out carefully and to a high technical standard. As alternatives to a true null-hypothesis (that phase dependent plasticity does not exist in the bottom-up processing stream for somatosensation), they propose that either the stimulation procedure was inadequate OR that the dependent measures were insensitive. The discussion of these possibilities and the results in general is thoughtful and sound. I would therefore recommend publication after the authors address the following

1.1. Thalamocortical inputs in primary sensory regions (from primary thalamic nuclei) primarily synapse in deeper layer 4, while the EEG signals and maximal TMS effects (from past studies) might be biased toward layer 1 of the cortex, where inputs from other cortical regions and secondary thalamic nuclei synapse. Perhaps the differences seen in this study compared to past findings of m1 phase-dependent plasticity/excitability reflect not a difference between motor and sensory cortices, but between bottom-up/lay4 vs top-down/intracortical/lay1 synapses.

Response: The scalp EEG is indeed most sensitive to cortical (rather than subcortical) sources and in particular to dipoles with radial orientation located on the gyral crown. However, despite the particular contribution of the most superficial cortical layers simply due to their shortest distance from the scalp, highly synchronized activity from deeper cortical layers can contribute significantly to the scalp surface signal as well via volume conduction. In fact, “due to their unique orientation with their long apical dendrites perpendicular to the cortical surface, large cortical pyramidal neurons in deep cortical layers play a major role in the generation of the EEG” (Kirschstein and Kohling, 2009). In particular for alpha oscillations, a relevant contribution was observed for layer 5 pyramidal neurons of the monkey (Bollimunta et al., 2011; Buffalo et al., 2011; Spaak et al., 2012; Haegens et al., 2015), and an independent origin in layer 5 neurons is also assumed specifically for the human mu-alpha rhythm (Silva et al., 1991; Jones et al., 2000; Jones et al., 2009). However, it is true that layer 4 neurons, where thalamocortical input synapses are located, may not produce a particularly strong contribution to the surface EEG signal. We still assume that a neural synchronization pattern as strong as the sensorimotor mu-alpha rhythm has an (indirect) impact on all layers of the somatosensory cortex, but possibly with a phase shift. We now discuss this issue more explicitly in the revised version of the paper.

P16: “Beside differences in preferred phase for phase-dependent plasticity between M1 and S1, gyral geometry and layer specific effects should be considered. It is possible that the effective phase at relevant thalamocortical synapses in layer 4 pyramidal cells of S1 is shifted relative to the scalp EEG, which is particularly sensitive to signals from superficial layers closest to the scalp but also highly synchronized signals originating from the long apical dendrites of large layer 5 pyramidal cells that are oriented perpendicular to the scalp at the gyral crown [31, 32], with the latter most strongly expressing the rhythms of the alpha band in the monkey [33-36] and likely also in the human brain [37-39]. Therefore, given the assumed location of phase-dependent synaptic plasticity in layer 4, the phase targets may have been suboptimal as well.”

1.2. The consideration of stimulation-related caveats is thorough (7 distinct points) but the discussion of limitations in their recording/analysis of the SEPs and STNs is lacking. I can think of several potential caveats -- some of which can be added to the discussion, while some may be directly explored.

Response: We thank the reviewer for pointing out this shortcoming and have now added a discussion of potential shortcomings in our ST and SEP analyses. For the reasoning behind this added section, please see the point-by-point responses to points 1.2.1-1.2.4 below.

P17: ”Ninth, our SEP and ST determination procedures may have been insensitive for the induced effects. Regarding the STs, we observed a general phase-condition unspecific increase over time, which may have been caused by sensory habituation or reduced attention but also a potential decrease in finger electrode impedance over time (which we did unfortunately not monitor), which may have in turn caused the constant current stimulator to apply lower voltages to achieve the fixed current strength, making the stimuli more difficult to detect. While considered unlikely, even a phase-unspecific general PES-intervention induced long term depression (LTD) is theoretically possible. However, irrespective of its reason, this general decrease cannot explain the lack of differential effects between the phase conditions. Regarding the SEPs, we have analyzed both carefully selected SEP ICA components from subject-specific SEP clusters as well as classical CP3-Fz-montage SEP components following minimalistic ICA cleaning and did not observe any differential effects in either one of them, neither with a frequentist nor a Bayesian statistical approach. While it remains possible that other, more derived EEG indices would have revealed unexpected differences elsewhere, we deliberately refrained from further exploratory analyses to prevent multiple comparison issues.”

1.2.1 The mu-rhythm was measured using a surface laplacian montage. This would likely bias the recorded Mu to superficial layers, while the relevant mu-alpha should be strongest in granular layer 4 (thalamocortical recipient layer). It has been shown that cortical regions may display distinct alpha-band oscillations, differentially related to attention and memory retention, and with one being potentially thalamocortical and another being corticocortical (see reviews by Klimesch et al as well as e.g. Lopes Da Silva, 1980; Halgren et al. 2019; Sokoliuk et al 2019; Rodriguez-Larios et al eNeuro). These different processes may differ in both frequency and spatial distribution. Further, there may be distinct visual vs somatosensory mu/alpha rhythms, which could both present in all electrodes (separable mainly via ICA) and therefore may have biased/affected the researchers ability to select the relevant mu/alpha rhythm. The chosen 8-14Hz range for identifying individual mu-alpha rhythms in this paper is a broad band, and it’s possible that the researchers selected the wrong oscillation to stimulate with respect to.

Response: We respectfully disagree with the interpretation that this montage would overly bias the extracted signal particularly to superficial cortical layers (see also our response to point 1.1). The specific spatial and frequency filters applied in this study were explicitly developed over many studies to successfully extract the local mu-alpha rhythm and disentangle it as good as possible from other rhythms of the alpha family. Given the large number of studies that successfully recorded the local mu-alpha rhythm with this specific extended Hjorth-style surface Laplacian montage (i.e., C3 referenced against the average of FC1, FC5, CP1, and CP5) (Thies et al., 2018; Zrenner et al., 2018; Bergmann et al., 2019; Hussain et al., 2019; Schaworonkow et al., 2019; Stefanou et al., 2019; Baur et al., 2020; Baur et al., 2022), we are highly confident that the stimulation was effectively triggered based on the phase of the local mu-alpha rhythm originating from the left somatosensory cortex (Zrenner et al., 2022). In fact, the used montage appears to be particularly effective in extracting the mu-alpha rhythm, even compared to more sophisticated spatial filter solutions (Schaworonkow et al., 2018). Also, we did in fact apply an individualized pass-band filter based on the individual mu-alpha frequency ± 2 Hz as described in the Methods section (page 8). As shown in previous publications (see e.g., Figure 4 in (Bergmann et al., 2019)), the topography of the respective mu-alpha power increases directly preceding the stimulation locates to the left sensorimotor region and not e.g., the visual cortex.

1.2.2. Is it possible that the sensory threshold and SEP tests induced some plasticity, making it more difficult to identify plasticity effects? upwards of 300 stimuli were applied to each finger before the intervention even began. The authors consider that the intervention (of 400 stimulations per finger) were not strong enough relative to the pre/post tests of SEP/ST, but what if the pre/post tests themselves were too strong?

Response: We agree with the reviewer and have extended the existing discussion point accordingly. 

P17: “Or, from a different perspective, maybe the many pulses applied during the SEP and ST measurements preceding the intervention did already induce some plastic after-effects that did block the subsequent attempt to induce plastic changes via the phase-triggered PES.”

1.2.3 The ICA-component selection methods seem highly subjective “Only components containing SEP information (i.e., resembling a typical N20-P25 waveform and a corresponding topography) were kept”. A more common approach in the literature is to use ICA only to remove irrelevant components (blinks, eye movements, musculature tension, cardiac signal). Could this metholodogy for isolating N20/P25 wavefoms have been too restrictive, leading to the null result?”

Response: Our approach of keeping only SEP-like ICA components was certainly conservative but not more subjective than other rule-driven ICA cleaning procedures removing components based on certain temporal and spatial features. However, we did also try the more conservative approach of extracting the N20 and P25 components from the classical CP3-Fz montage after minimalistic ICA cleaning (i.e., removing only eye blink/movement components), which did not reveal any positive results either. We have now mentioned this approach in the Methods and the negative findings in the Results section of the revised manuscript for sake of completeness. 

P9/10: “In addition, N20 and P25 amplitudes were also extracted from the classical CP3-Fz montage after minimal ICA cleaning (only removing obvious eye blink/movement components; removed components per subject: 1.53 ± 0.62, mean ± SD).”

P11: “In addition, SEPs extracted from the classical CP3-Fz montage after minimal ICA cleaning revealed no significant main effects of Phase or Time or their interaction for any of the analyses described above (all p > 0.2; BF01 > 2). “

1.2.4 On the other hand, because only the p20/p25 components were retained after the ICA, could the authors use a global-field power estimate of the ERP amplitude? this might be a more powerful/sensitive metric than simply using the same limited cluster of electrodes across all subjects.

Response: We deliberately did not use the same cluster of electrodes across all subjects but selected them individually to account for the different topographies across subjects (which the classical CP3-Fz montage does not). Based on our hypothesis, we were specifically looking for SEP changes in the somatosensory cortex, which is well reflected in this specific cluster, and we prefer not to add additional exploratory analyses without spatial specificity such as the GMFP.

1.3. Further, was the impedance of the finger electrodes tested before and after the intervention?

Response: We thank the reviewer for raising this point. We did not measure the impedance of the finger electrodes before and after the intervention. While this may possibly explain the general increase of sensory thresholds over time: theoretically, with decreasing impedances, the constant current stimulator would have to apply lower voltages to produce the fixed current strength and stimuli would thus become more difficult to detect). While this is a methodological shortcoming we will definitely fix in future studies of this kind, such a general impedance change over time would affect all conditions similarly and not explain and differences between conditions - nor the lack of lack thereof as in our case. We have added these considerations to the Discussion.

P17: ”Regarding the STs, we observed a general phase-condition unspecific increase over time, which may have been caused by sensory habituation or reduced attention but also a potential decrease in finger electrode impedance over time (which we did unfortunately not monitor), which may have in turn caused the constant current stimulator to apply lower voltages to achieve the fixed current strength, making the stimuli more difficult to detect. While considered unlikely, even a phase-unspecific general PES-intervention induced long term depression (LTD) is theoretically possible. However, irrespective of its reason, this general decrease cannot explain the lack of differential effects between the phase conditions.”

1.4. it is hard to discern different conditions in the line plots in figure 2A, although I understand that this is, to some degree, the point! nevertheless, please spread out the x-axis or make the lines thinner.

Response: We have now modified Figure 2 A accordingly, making the lines slightly thinner. However, as the reviewer indicated already, a certain overlap is indeed simply owed to the lack of differences.

1.5. Authors should discuss the absence of a “no stimulation” or sham condition.

Response: We do now discuss the reasoning behind including a random phase condition instead of a no-stimulation condition in the Discussion.

P17: “Finally, we did not include a low-level control condition such as “sham” or no-stimulation, i.e., a finger to which no effective PES would be applied in between SEP and ST measures, but only a high-level control condition with random-phase PES application. While a no-stimulation condition may have helped to explain the general ST increases, only the random condition did effectively control for any placebo effects or other nonspecific stimulation effects that are not due to the phase-dependent plasticity mechanism under investigation.”

1.6. I may be confused, but the fact that some INCREASE in sensory threshold was found for all cases is interesting. This is a drop in sensitivity. It is difficult to make sense of this effect in light of their interpretation that their protocol should induce LTP but not induce LTD (which, it may have in all three cases). This effect was strong and clearly present in nearly all subjects (and was statistically significant). Could a change in stimulation electrode impedance/contact/etc account for it? Or changes in the tissue e.g. inflammation? Or perhaps changes in arousal could have lead to drops in skin conductance, which would alter the impact of the stimulating electrode.

Response: We have now elaborated on this issue in the Discussion (see also point 1.3).

P17: “Regarding the STs, we observed a general phase-condition unspecific increase over time, which may have been caused by sensory habituation or reduced attention but also a potential decrease in finger electrode impedance over time (which we did unfortunately not monitor), which may have in turn caused the constant current stimulator to apply lower voltages to achieve the fixed current strength, making the stimuli more difficult to detect. While considered unlikely, even a phase-unspecific general PES-intervention induced long term depression (LTD) is theoretically possible. However, irrespective of its reason, this general decrease cannot explain the lack of differential effects between the phase conditions.”

1.7. Another potential explanation for the null results could have been that plasticity in synapses upstream from thalamocortical inputs to S1 masked, blunted, or compensated for thalamocortical plasticity (in a phase-dependent or independent manner). Indeed, as they point out in the discussion, a past study identified only plasticity for concurrent PES and direct cortical TMS stimulation, and this affected mainly the p25, interpreted as a layer 1 effect, while the n20 was not affected (interpreted as a lack of a thalamocortical effect).

Response: We have extended an existing discussion point to include this notion.

P17: “Eights, it is possible that phase-dependent plastic changes were indeed induced somewhere in S1, but not at those neuronal elements that would result in a change of SEP N20 or P25 amplitude or sensory threshold. Alternatively, plasticity induced in relevant thalamic-cortical synapses may have been masked or compensated for by other plastic changes in synapses upstream to the thalamocortical input layer.”

Reviewer #2: Thank you for the opportunity to review this manuscript.

The experiment investigates whether (and how) sensorimotor mu-alpha phase alters cortical excitability in sensory cortex. The authors use state of the art real-time EEG phase prediction to trigger peripheral electrical stimulation timed to occur at different parts of the sensorimotor mu oscillation. The goal is to induce phase-dependent plasticity in thalamocortical synapses, as has been shown in corticospinal output from M1. However there were no differences observed between the peripheral pulses timed to occur in peak, trough or random phases.

2.1 This is a negative study, however as outlined in the discussion, there are several key limitations to the experimental design that may have undermined the ability to draw any conclusions about the proposed relationship between phase and excitability. Most importantly, it is not clear that the phase estimated at C3 would be the same as phase overlying the sensorimotor cortex. As the authors mention, even a small difference in electrode placement and orientation can cause a significant shift in the measured oscillatory phase. This issue may mean that while the estimation is reliable at C3, the only conclusion that can be drawn is that SEPs delivered according to mu-alpha phase measured at C3 do not affect plasticity in nearby sensory cortex.

Response: While there is no definitive certainty regarding the specific phase relationship between the mu-alpha rhythm extracted via the extended C3-centered Hjorth montage and the relevant neuronal elements in the primary somatosensory cortex, recent evidence is at least emphasizing that this particular montage is particularly reflecting post-central somatosensory rather than precentral motor cortical mu-alpha signals (Zrenner et al., 2022). Moreover, this particular montage and the particular phase angles targeted were identified in previous work to be well suited for revealing phase-dependent effects on corticospinal plasticity (Schaworonkow et al., 2018) and have thus been used in a large number of papers (Thies et al., 2018; Zrenner et al., 2018; Bergmann et al., 2019; Hussain et al., 2019; Schaworonkow et al., 2019; Stefanou et al., 2019; Baur et al., 2020; Baur et al., 2022), highlighting their practical relevance for the research field of real-time EEG-triggered stimulation. However, we cannot rule out that a slightly different montage or slightly other target phase angles may have produced positive effects, and we have discussed this extensively in the first discussion point of the respective section on page 15/16. 

All other limitations as discussed by the authors may also contribute to the negative findings. In addition, the choice to have the central pulse of a triplet at the target (i.e., have one pulse just before, one at target and one after target) may also play a role as some evidence suggests that rising or falling phase is important for long term potentiation, rather than peak or trough itself.

Response: This is indeed another important aspect, and we have now extended our second discussion point on page 15/16 accordingly.

P16: “Second, on the other hand, using PES triplets instead of single pulses may have also led to a reduction of temporal precision and prevented the proposed mechanisms of phase-dependent plasticity to take effect. However, given the effective induction of LTP-like plasticity using mu-alpha trough targeted 100 Hz TMS triplets [6] as well as the successful use of high-frequency bursts in a novel paired associative stimulation (PAS) protocol [40, 41], this choice is unlikely responsible for the complete lack of phase-specific after-effects observable in this study. Nonetheless, it should be noted in that previous work using 100 Hz TMS triplets, the first of the three pulses pulse was applied at the target phase angle while the following two pulses already fell into the beginning of the rising phase of the mu-alpha cycle, a phase that was later shown to be even more excitable than the very trough [42]. It is thus possible that a slightly different phase-timing of the triplet would have led to different results in the current study.”

Minor points:

2.2

The channels listed in figure 2B overlap and are unreadable.

Response: We have now removed the unreadable channel names in Figure 2B and indicated channel positions with dots only. 

2.3

The boxplots with 25-75% ranges within the violin plots are so small

that they are unreadable in figure 2 and 3.

Response: We have now increased the size of the embedded box plots in the violin plots in Figure 2 and Figure 3.

References

Baur D, Ermolova M, Souza VH, Zrenner C, Ziemann U (2022) Phase-amplitude coupling in high-gamma frequency range induces LTP-like plasticity in human motor cortex: EEG-TMS evidence. Brain Stimul.

Baur D, Galevska D, Hussain S, Cohen LG, Ziemann U, Zrenner C (2020) Induction of LTD-like corticospinal plasticity by low-frequency rTMS depends on pre-stimulus phase of sensorimotor mu-rhythm. Brain Stimul 13:1580-1587.

Bergmann TO, Lieb A, Zrenner C, Ziemann U (2019) Pulsed Facilitation of Corticospinal Excitability by the Sensorimotor mu-Alpha Rhythm. J Neurosci 39:10034-10043.

Bollimunta A, Mo J, Schroeder CE, Ding M (2011) Neuronal mechanisms and attentional modulation of corticothalamic alpha oscillations. J Neurosci 31:4935-4943.

Buffalo EA, Fries P, Landman R, Buschman TJ, Desimone R (2011) Laminar differences in gamma and alpha coherence in the ventral stream. Proc Natl Acad Sci U S A 108:11262-11267.

Haegens S, Barczak A, Musacchia G, Lipton ML, Mehta AD, Lakatos P, Schroeder CE (2015) Laminar Profile and Physiology of the alpha Rhythm in Primary Visual, Auditory, and Somatosensory Regions of Neocortex. J Neurosci 35:14341-14352.

Hussain SJ, Claudino L, Bonstrup M, Norato G, Cruciani G, Thompson R, Zrenner C, Ziemann U, Buch E, Cohen LG (2019) Sensorimotor Oscillatory Phase-Power Interaction Gates Resting Human Corticospinal Output. Cereb Cortex 29:3766-3777.

Jones SR, Pinto DJ, Kaper TJ, Kopell N (2000) Alpha-frequency rhythms desynchronize over long cortical distances: a modeling study. J Comput Neurosci 9:271-291.

Jones SR, Pritchett DL, Sikora MA, Stufflebeam SM, Hamalainen M, Moore CI (2009) Quantitative analysis and biophysically realistic neural modeling of the MEG mu rhythm: rhythmogenesis and modulation of sensory-evoked responses. J Neurophysiol 102:3554-3572.

Kirschstein T, Kohling R (2009) What is the source of the EEG? Clin EEG Neurosci 40:146-149.

Schaworonkow N, Triesch J, Ziemann U, Zrenner C (2019) EEG-triggered TMS reveals stronger brain state-dependent modulation of motor evoked potentials at weaker stimulation intensities. Brain Stimul 12:110-118.

Schaworonkow N, Caldana Gordon P, Belardinelli P, Ziemann U, Bergmann TO, Zrenner C (2018) mu-Rhythm Extracted With Personalized EEG Filters Correlates With Corticospinal Excitability in Real-Time Phase-Triggered EEG-TMS. Front Neurosci 12:954.

Silva LR, Amitai Y, Connors BW (1991) Intrinsic oscillations of neocortex generated by layer 5 pyramidal neurons. Science 251:432-435.

Spaak E, Bonnefond M, Maier A, Leopold DA, Jensen O (2012) Layer-specific entrainment of gamma-band neural activity by the alpha rhythm in monkey visual cortex. Curr Biol 22:2313-2318.

Stefanou MI, Baur D, Belardinelli P, Bergmann TO, Blum C, Gordon PC, Nieminen JO, Zrenner B, Ziemann U, Zrenner C (2019) Brain State-dependent Brain Stimulation with Real-time Electroencephalography-Triggered Transcranial Magnetic Stimulation. J Vis Exp.

Thies M, Zrenner C, Ziemann U, Bergmann TO (2018) Sensorimotor mu-alpha power is positively related to corticospinal excitability. Brain Stimul 11:1119-1122.

Zrenner C, Desideri D, Belardinelli P, Ziemann U (2018) Real-time EEG-defined excitability states determine efficacy of TMS-induced plasticity in human motor cortex. Brain Stimul 11:374-389.

Zrenner C, Belardinelli P, Ermolova M, Gordon PC, Stenroos M, Zrenner B, Ziemann U (2022) micro-rhythm phase from somatosensory but not motor cortex correlates with corticospinal excitability in EEG-triggered TMS. J Neurosci Methods 379:109662.

---

## [Decision Letter · Decision Letter 1]

16 Oct 2023

Repetitive sensorimotor mu-alpha phase-targeted afferent stimulation produces no phase-dependent plasticity related changes in somatosensory evoked potentials or sensory thresholds

PONE-D-23-15283R1

Dear Dr. Bergmann,

We’re pleased to inform you that your manuscript has been judged scientifically suitable for publication and will be formally accepted for publication once it meets all outstanding technical requirements.

Kind regards,

Gennady S. Cymbalyuk, Ph.D.

Academic Editor

PLOS ONE

Additional Editor Comments (optional):

Reviewers' comments:

Reviewer's Responses to Questions

**Comments to the Author**

1. If the authors have adequately addressed your comments raised in a previous round of review and you feel that this manuscript is now acceptable for publication, you may indicate that here to bypass the “Comments to the Author” section, enter your conflict of interest statement in the “Confidential to Editor” section, and submit your "Accept" recommendation.

Reviewer #1: All comments have been addressed

2. Is the manuscript technically sound, and do the data support the conclusions?

Reviewer #1: Yes

3. Has the statistical analysis been performed appropriately and rigorously? 

Reviewer #1: Yes

4. Have the authors made all data underlying the findings in their manuscript fully available?

Reviewer #1: Yes

5. Is the manuscript presented in an intelligible fashion and written in standard English?

Reviewer #1: Yes

6. Review Comments to the Author

Reviewer #1: The authors have thoroughly and adequately responded to my concerns and adjusted their discussion accordingly. It is publishable in its current form.

As a final note, I might be misunderstanding something, but I still might suspect that taking the global field power on the ICA-extracted p20/25 components -or taking some spatial composite or component score- might prove more powerful in estimating small changes in these ERPs than restricting to just a few electrodes, as a) ERPs are volume conducted anyway and typically extend scalp-wide and b) the ICA will apply spatial weights, such that some useful data can be gleaned from all electrodes in this case.

That said, the methodology is clearly stated. As written, this paper will be useful in guiding future studies on what might and might not work.

7. PLOS authors have the option to publish the peer review history of their article (what does this mean?). If published, this will include your full peer review and any attached files.

Reviewer #1: No

---

## [Editor Report · Acceptance letter]

19 Oct 2023

PONE-D-23-15283R1 

Repetitive sensorimotor mu-alpha phase-targeted afferent stimulation produces no phase-dependent plasticity related changes in somatosensory evoked potentials or sensory thresholds 

Dear Dr. Bergmann:

I'm pleased to inform you that your manuscript has been deemed suitable for publication in PLOS ONE. Congratulations! Your manuscript is now with our production department. 

Kind regards, 

on behalf of

Dr. Gennady S. Cymbalyuk 

Academic Editor

PLOS ONE